# Adsorption Kinetics Model of Hydrogen on Graphite

**DOI:** 10.3390/e27030229

**Published:** 2025-02-23

**Authors:** Jean-Marc Simon, Guilherme Carneiro Queiroz da Silva

**Affiliations:** Laboratoire Interdisciplinaire Carnot de Bourgogne, UMR-6303 CNRS-Université Bourgogne Europe (UBE), 9 Av. A. Savary, 21000 Dijon, France; gcarneiroq@gmail.com

**Keywords:** adsorption kinetics model, hydrogen on graphite, transport equation, transport coefficients

## Abstract

A new kinetic equation for the adsorption and desorption of H_2_ on graphite is derived based on the adsorption and desorption equilibrium rates obtained from the molecular dynamics. These rates are proportional to the activity in the gas and the adsorbed phase and thus do not obey Langmuir kinetics. The new equation offers a new route for understanding experimental results. It is used to simulate the kinetics under different thermodynamic conditions, both isothermal and non-isothermal. The characteristic times of adsorption and desorption are in good agreement with the data from the literature. The relation between the kinetics and the mass flow equation is discussed within the framework of the non-equilibrium thermodynamics of heterogeneous systems. Finally, expressions for the transport coefficients are proposed for both the transfer of mass and the coupling between the mass and heat fluxes.

## 1. Introduction

Hydrogen-based technologies are an alternative for reducing the production of greenhouse gases like CO_2_. The current research aims to optimise the handling of this resource under safe and economically viable conditions [1,2,3]. Its production through electrolysis as a solution for storing energy and the recent discovery of geosourced hydrogen [4,5,6] have generated growing interest in its recovery and valorisation. However, a better understanding of the physical chemistry of hydrogen is essential to addressing challenges related to its production, storage, transport, and final use.

The traditional methods of storage, like mechanical compression at high pressure or liquefaction at cryogenic temperatures, though effective, must overcome challenges related to safety, corrosion, and energy costs. New solutions are exploring the interaction between hydrogen and different types of materials [1,2,3]. It can simply be adsorbed onto adsorbent materials such as nanostructured carbon materials, Metal– and Covalent Organic Frameworks (MOFs and COFs), polymers, or zeolites. These materials are generally characterised by their safe and good reversibility in use (adsorption/desorption) and can offer high storage capacities. For the development of safe and efficient conditions of storage, it is important to consider not only the thermodynamics of adsorption but its kinetics as well.

In order to investigate the interplay between the thermodynamics and dynamics in physical adsorption processes, Simon et al., in a previous paper [7], studied a simple system using molecular dynamics: a gas phase of di-hydrogen in contact with a graphite surface. Equilibrium states were simulated at temperatures ranging from 70 K to 390 K using different amounts of hydrogen molecules. The population of adsorbed molecules was recorded, and the thermodynamic isotherms were computed. The results showed good agreement with Langmuir’s adsorption isotherm:(1)$$\theta_{eq}=\frac{Ka^{g}}{1+Ka^{g}},\;\mathrm{with}\;a^{g}=p/p^{0}.$$
where $a^{g}$, *p*, $p^{0}$, and $\theta_{eq}$ are the activity of the gas phase and its pressure, normal pressure (1 bar), and the coverage under equilibrium conditions. The definition of θ in the equations above follows Langmuir’s original definition, which is the fractional coverage of a monolayer. In other words, this is the ratio between the surface concentration, $c^{s}$, and the saturated surface concentration, $c^{sat}$. *K* is Langmuir’s constant. Langmuir’s model is based on non-interacting and non-mobile adsorbed molecules, while the simulations clearly evidence that the hydrogen molecules are mobile and interact between themselves. An interesting discussion on the interpretation of the ideality or non-ideality of the adsorbed phase in terms of the interaction between the adsorbed molecules is proposed in [8] based on the results [7]. The qualitative results we present here do not change with the choice of one interpretation over another; however, quantitative differences may appear.

Following [7], the adsorbed phase shows non-ideal behaviour that falls outside of Henry’s regime (at low coverage), which is revealed in the expression of the chemical potential:(2)$$\mu^{ads}=\mu^{★}+RT\ln a^{ads},\;\mathrm{with}\;a^{ads}=\frac{\theta }{1-\theta }=\gamma \theta .$$
Here, *T* and *R* are the temperature (K) and the perfect gas constant, and $\mu^{★}$ is the standard state in Henry’s law. $a^{ads}$ is the chemical activity of the adsorbed phase, and γ=1/(1−θ) is the activity coefficient. Outside of Henry’s regime (θ≈0), γ deviates from one, which is a sign of non-ideal thermodynamic behaviour.

The condition of equilibrium demands that the chemical potential of the adsorbed phase $\mu^{ads}$ and that of the gas phase $\mu^{g}$ are equal. Using $\mu^{0}$, the standard state of the gas phase, the chemical potential of the gas phase is(3)$$\mu^{g}=\mu^{0}+RT\ln a^{g},$$In the same molecular dynamics study, Simon et al. [7] computed the unidirectional rates of adsorption and desorption under equilibrium conditions. They proposed new kinetic equations for the adsorption/desorption of mobile hydrogen molecules that did not agree with the non-mobile model of Langmuir kinetics. The rates, $J_{a}$ (adsorption) and $J_{d}$ (desorption), are shown to be proportional to the thermodynamic activity of the gas, $a^{g}$, and of the adsorbed phase, $a^{ads}$: (4)$$\begin{array}{ccc} J_{a} & = & k_{a}a^{g}=k_{a}p/p^{0}, \end{array}$$(5)$$\begin{array}{ccc} J_{d} & = & k_{d}a^{ads}=k_{d}\frac{\theta }{1-\theta }. \end{array}$$The coefficients $k_{a}$ and $k_{d}$ are the adsorption and desorption rate constants, respectively. Using Equations (Equation 4) and (Equation 5) with $J_{a}=J_{d}$, the equilibrium conditions, we obtain the adsorption Langmuir equation:(6)$$k_{a}\frac{p}{p^{0}}=k_{d}\frac{\theta_{eq}}{1-\theta_{eq}}.$$Compared with Equation (Equation 1), the Langmuir constant is given by the ratio $k_{a}/k_{d}=K$.

For the adsorption of hydrogen on graphite, it was reported [7] that both the rate constant and the saturated surface concentration follow an Arrhenius law: (7)$$\begin{array}{ccc} c^{sat} & = & 1.35\times 10^{-5}\exp\left(\frac{180}{RT}\right), \end{array}$$(8)$$\begin{array}{ccc} k_{a} & = & 3.5\times 10^{3}\exp\left(\frac{1500}{RT}\right), \end{array}$$(9)$$\begin{array}{ccc} k_{d} & = & 1.8\times 10^{7}\exp\left(\frac{-4400}{RT}\right). \end{array}$$
The units of the activation energy and of the constant are, respectively, J/mol and mol/m^2^s, while $c^{sat}$ is given in mol/m^2^.

The equations are consistent with the thermodynamics of a Langmuir process (isothermal adsorption) but, as mentioned above, not with Langmuir kinetics: (10)$$\begin{array}{ccc} J_{a}^{L} & = & k_{a}^{L}p\left(1-\theta \right), \end{array}$$(11)$$\begin{array}{ccc} J_{d}^{L} & = & k_{d}^{L}\theta . \end{array}$$
Here, $J_{a}^{L}$, $J_{d}^{L}$, $k_{a}^{L}$, and $k_{d}^{L}$ are the Langmuir adsorption and desorption fluxes and rate constants.

It is worth noticing that for a low coverage limit, in Henry’s regime with 1−θ=1, both the Langmuir and the proposed kinetic models lead to the same expressions for the fluxes: (12)$$\begin{array}{ccc} J_{a}^{H} & = & k_{a}^{H}p, \end{array}$$(13)$$\begin{array}{ccc} J_{d}^{H} & = & k_{d}^{H}\theta , \end{array}$$
where $J_{a}^{H}$, $J_{d}^{H}$, $k_{a}^{H}$, and $k_{d}^{H}$ are Henry’s adsorption and desorption fluxes and rate constants, respectively. So, when 1−θ≈1, we have the following: (14)$$\begin{array}{c} k_{a}^{H}=k_{a}=k_{a}^{L}, \end{array}$$(15)$$\begin{array}{c} k_{d}^{H}=k_{d}=k_{d}^{L}\theta . \end{array}$$One particular feature of the proposed model is that it takes into account the non-ideality of the adsorbed phase. This is facilitated by including the dependence of the desorption rate on the activity coefficient (γ). The new expressions therefore apply to non-ideal mobile adsorbed phases.

The aim of this work is to develop a set of equations that are able to describe the kinetics of the adsorption/desorption of H_2_ on graphite under isothermal and non-isothermal conditions. This paper is divided as follows: First, theoretical considerations comparing the proposed model to the classical Langmuir expressions are given. Then, the link between the new model and transport equations are discussed. In the last part, the numerical results obtained using the new equations are also shown.

## 2. Theoretical Considerations

### 2.1. Isothermal Adsorption Kinetics

Common to all reaction schemes is the general expression for the net rate of adsorption, $J_{net}$:(16)$$J_{net}=c^{sat}\frac{d\theta (t)}{dt}=J_{a}-J_{d},$$
where *t* is the time. In this expression, the positive flux is defined as the mass transfer from the gas to the adsorbed phase. Considering the expression for $J_{a}$ and $J_{d}$, Equations (Equation 4) and (5), and (Equation 10)–(13), the net rate is therefore a function of p(t) and θ(t), which are functions of time. For simplicity, the equations are written without explicit temporal dependence.

Relevant adsorption experiments are often conducted under isobaric conditions. Solving the differential equations at a constant *p* results in analytical expressions for the adsorption rate curves. Non-isobaric conditions lead to more complex solutions that can be obtained through numerical integration. This latter type will not be considered here.

At this point, it is necessary to discuss whether the expressions of $J_{a}$ and $J_{d}$ are valid outside of equilibrium conditions. These expressions assume microscopic reversibility and thus hold close to equilibrium. Far from equilibrium, the two models, Langmuir and non-Langmuir, can be considered as a first approximation. Nonetheless, the results should be regarded with caution. Conditions far from equilibrium will be used to illustrate the behaviour predicted by the mathematical expressions obtained for the mass flux according to the new model.

The flux equations defined in [7] involve one additional assumption that is based on consideration of the Langmuir kinetics. From the solid to the gas phase, three zones are considered. The first is the adsorbed phase, where the molecules are in contact with the solid surface. The second is a surface buffer region a few angstroms thick, where the molecules do not behave in the same way as they do in the bulk/gas phase. In other words, their behaviour is still driven by interactions with the surface. Where H_2_ is close to the graphite surface, the adsorbate molecules are more densely packed due to the attractive forces between molecules and both the solid and already adsorbed H_2_. The last region is the gas phase, where the effects of the interface are negligible. Note that the gas and solid phases are not in direct contact in this framework. This segmentation was used in ref. [7] to compute the fluxes. $J_{a}$ was calculated as the part of the flux that reached the adsorbed surface area originating from the gas phase. $J_{d}$ was the part of the flux that reached the gas phase and came from the adsorbed phase. Thus, these two fluxes are not located at the same plane positions. The definition of the net flux, Equation (Equation 16), is the transport of mass across this buffer surface region, and the net flux is, on average, assumed to be constant through it. It is necessary to stress that this buffer zone is not explicitly analysed, but its role in the adsorption kinetics is taken into account through the values of the rate constants in the kinetics, that is, $k_{a}$ and $k_{d}$. A discussion on the link between the Langmuir and non-Langmuir kinetics and the role of correlations across an equivalent of the surface buffer is proposed in [8].

Below, we present equations for the temporal evolution of the coverage, θ. This set of equations is the result of integrating the expressions present in the kinetic models just discussed (Equations (Equation 4), (5), (Equation 10), and (11)). Henry’s regime is also present to give a limited ideal case as a reference for model comparison.

#### 2.1.1. Net Rate—Non-Langmuir Kinetics

Equation (Equation 16) is integrated for non-ideal surface kinetics using Equations (Equation 4) and (5). Introducing the expressions of the rates, Equations (Equation 4) and (5), into Equation (Equation 16) results in the following:(17)$$c^{sat}\frac{d\theta }{dt}=k_{a}\frac{p}{p^{0}}-k_{d}\frac{\theta }{1-\theta },$$
or(18)$$c^{sat}\left(1-\theta \right)\frac{d(1-\theta )}{dt}=-\left(1-\theta \right)\left(k_{a}\frac{p}{p^{0}}+k_{d}\right)+k_{d},$$
which can be written as(19)$$c^{sat}\left(\frac{X}{k_{d}-AX}\right)dX=dt,$$
with X=(1−θ) and $A=\left(k_{a}\frac{p}{p^{0}}+k_{d}\right)$.

We integrate this expression from the initial coverage at time zero, $\theta_{i}$ ($X_{i}=X\left(t=0\right)$), into the coverage at time *t* and obtain the following:(20)$$\frac{c^{sat}}{A}\left[X_{i}-X+\frac{k_{d}}{A}\ln\left(\frac{k_{d}-AX_{i}}{k_{d}-AX}\right)\right]=t,$$
or(21)$$\frac{c^{sat}}{A}\left[\theta -\theta_{i}+\frac{k_{d}}{A}\ln\left(\frac{\theta_{i}A-k_{a}\frac{p}{p^{0}}}{\theta A-k_{a}\frac{p}{p^{0}}}\right)\right]=t,$$
where *A* is a constant under isobaric conditions. *A* can simply be obtained from the equilibrium isotherm, Equation (Equation 6):(22)$$A=k_{a}\frac{p}{p^{0}}+k_{d}=k_{d}\left(\frac{1}{1-\theta_{eq}}\right).$$By introducing the expression for *A* into Equation (Equation 21), we can write *t* as a function of θ:(23)$$\frac{c^{sat}{\left(1-\theta_{eq}\right)}^{2}}{k_{d}}\left[\frac{\theta -\theta_{i}}{1-\theta_{eq}}+\ln\left(\frac{\theta_{i}-\theta_{eq}}{\theta -\theta_{eq}}\right)\right]=t.$$Equation (Equation 23) has a well-defined solution for θ as a function of *t*, even if we are close to saturation (full coverage). When $\theta_{eq}$ is close to unity, the first term in the square brackets dominates at a low value of *t*, making θ proportional to *t*. The logarithm term dominates in most cases, when θ approaches equilibrium or a high value of *t*.

Full coverage at $\theta_{eq}=1$ can be understood as the case of an infinitely large pressure and an infinitely fast adsorption rate, with t=0. This limit cannot be reached for Equation (Equation 23) or for Langmuir kinetics; see Equation (Equation 27) below. Desorption towards $\theta_{eq}$=0 (zero pressure) is, on the other hand, well described by Equation (Equation 23).

The expressions for the characteristic times for adsorption, $t_{ads}$, and desorption, $t_{des}$, are the same, as expected for a process which obeys microscopic reversibility. They can be studied analytically from the long-term logarithmic behaviour of Equation (Equation 23):(24)$$t_{ads}=t_{des}=\frac{c^{sat}{\left(1-\theta_{eq}\right)}^{2}}{k_{d}}=\frac{c^{sat}}{k_{d}{\left(\gamma_{\theta_{eq}}\right)}^{2}}.$$The characteristic time is a function of $\theta_{eq}$ and not of $\theta_{i}$. This will lead to asymmetry in the time dependence of adsorption vs. desorption. However, Equation (Equation 23) involves linear dependence in addition to logarithmic behaviour, so it may be useful to define another characteristic time τ corresponding to a progress of 100(1−1/e)≈63%, as for pure logarithmic behaviour (see Langmuir’s kinetics and Henry’s regime):(25)$$\tau =\frac{c^{sat}{\left(1-\theta_{eq}\right)}^{2}}{k_{d}}\left[1+\frac{\left(1-1/e\right)\left(\theta_{eq}-\theta_{i}\right)}{\left(1-\theta_{eq}\right)}\right].$$It is worth noting that τ is a function of both the final coverage and the initial coverage.

#### 2.1.2. Net Rate—Langmuir Kinetics

For comparison, the expressions obtained from Langmuir kinetics (Equations (Equation 10) and (11)) are also given. By inserting the expressions of the rates, Equations (Equation 10) and (11), into Equation (Equation 16), we obtain the following:(26)$$c^{sat}\frac{d\theta }{dt}=k_{a}^{L}\frac{p}{p^{0}}\left(1-\theta \right)-k_{d}^{L}\theta .$$Integrating this equation with time leads to the evolution of the surface coverage.(27)$$\theta \left(t\right)=\theta_{eq}-\left(\theta_{eq}-\theta_{i}\right)\exp\left[-\frac{k_{d}^{L}t}{c^{sat}\left(1-\theta_{eq}\right)}\right],\;\;\;\;t_{ads}^{L}=t_{des}^{L}=\frac{c^{sat}\left(1-\theta_{eq}\right)}{k_{d}^{L}},$$
where $t^{L}$ is the characteristic time of the adsorption/desorption kinetics from Langmuir expressions. Under the specific equilibrium conditions and using the flux desorption rate expressions, Equations (5) and (11), it follows that $k_{d}^{L}=k_{d}/\left(1-\theta_{eq}\right)$, and the expressions for the characteristic times for Langmuir and non-Langmuir kinetics are then identical $t^{L}=t_{ads}$, but they differ from τ.

#### 2.1.3. Net Rate—Henry’s Regime

Under Henry’s regime, the interaction between the adsorbed particles is neglected, and the activity coefficient γ=1−θ is approximated to 1. Using Equations (Equation 12), (13), (Equation 15), and (16), we obtain the following:(28)$$c^{sat}\frac{d\theta }{dt}=k_{a}\frac{p}{p^{0}}-k_{d}\theta .$$Through integration, we obtain the following:(29)$$\theta \left(t\right)=\theta_{eq}-\left(\theta_{eq}-\theta_{i}\right)\exp\left(-\frac{k_{d}t}{c^{sat}}\right),t_{ads}=t_{des}=\frac{c^{sat}}{k_{d}},$$

### 2.2. The Transport Equation

Close to equilibrium, in the linear regime, the transport of heat and mass obeys the phenomenological equations described in the book by S.R. de Groot and P. Mazur [9] on homogeneous systems and in S. Kjelstrup and D. Bedeaux’s book [10] on heterogeneous systems. Based on the expression of the entropy production, the fluxes are written as a product of the transport coefficients and thermodynamic forces. In S. Kjelstrup and D. Bedeaux’s monograph [10], the transport equations are alternatively given in terms of resistivity coefficients. When we consider a surface, the thermodynamic driving forces are scalar properties and are expressed as the difference in the thermodynamic properties across the surface. Because of the heterogenous nature of the surface, the fluxes are different across the surface, except under stationary conditions. It is then useful to express the transport equations, i.e., the fluxes and transport coefficients, taking two different types of references: the gas side and the adsorbed phase side of the surface buffer, $J^{g}$ and $J^{ads}$, respectively. The two references are equivalent and can be used interchangeably depending on the practical criteria. For non-isothermal conditions, the mass flux expressions are as follows: (30)$$\begin{array}{ccc} J^{g} & = & -L_{\mu q}^{g}\left(\frac{1}{T^{g}}-\frac{1}{T^{ads}}\right)+L_{\mu \mu }^{g}\left(\frac{\mu^{g}\left(T^{g}\right)-\mu^{ads}\left(T^{g}\right)}{T^{g}}\right), \end{array}$$(31)$$\begin{array}{ccc} J^{ads} & = & -L_{\mu q}^{ads}\left(\frac{1}{T^{g}}-\frac{1}{T^{ads}}\right)+L_{\mu \mu }^{ads}\left(\frac{\mu^{g}\left(T^{ads}\right)-\mu^{ads}\left(T^{ads}\right)}{T^{ads}}\right), \end{array}$$
where $L_{\mu \mu }$ and $L_{\mu q}$ are, respectively, the mass and the cross-coupling (heat and mass) transport coefficients. Clearly, $J^{g}$ and $J^{ads}$ are different except under a stationary state, when the mass flux is constant across the whole surface. In our model, as mentioned above, the net flux, Equation (Equation 16), is assumed to be constant, and then $J^{g}=J^{ads}=J_{net}$. In the following, the mass transfer equation is derived based on Equation (Equation 16) for the net flux and using the non-Langmuir kinetics for both isothermal and non-isothermal conditions.

#### 2.2.1. Isothermal Conditions

Under a constant pressure, the activity of the gas phase is constant. At equilibrium $J_{net}=0$, $k_{a}a^{g}=k_{d}a_{eq}^{ads}$. This is used to defined two sets of transport equations, like for Equations (Equation 30) and (31). From Equations (Equation 4), (5), and (Equation 16), we obtain the following: (32)$$\begin{array}{ccc} J_{net} & = & k_{d}\left(a_{eq}^{ads}-a^{ads}\right)=k_{a}\left(a^{g}-a^{g\phi }\right), \end{array}$$(33)$$\begin{array}{ccc} J_{net} & = & k_{d}\exp\left(\frac{-\mu^{★}+\mu_{eq}^{ads}}{RT}\right)\left[1-\exp\left(\frac{\mu^{ads}-\mu_{eq}^{ads}}{RT}\right)\right], \end{array}$$(34)$$\begin{array}{ccc} J_{net} & = & k_{a}\exp\left(\frac{-\mu^{0}+\mu^{g}}{RT}\right)\left[1-\exp\left(\frac{\mu^{g\phi }-\mu^{g}}{RT}\right)\right], \end{array}$$The superscript ϕ for the gas indicates a state of equilibrium with the adsorbed phase $\mu^{ads}=\mu^{g\phi }$. The related activity, $a^{g\phi }$, is simply given by the Langmuir equation, Equation (Equation 1). Close to equilibrium, $\mu_{eq}^{ads}\approx \mu^{ads}$ or $\mu^{g}\approx \mu^{g\phi }$. The exponential function can then be expanded into a series, and the above transport equations can be written as the difference in the chemical potential across the surface, in agreement with [10]: (35)$$\begin{array}{ccc} J_{net} & = & \frac{k_{d}}{R}a_{eq}^{ads}\left(\frac{\mu_{eq}^{ads}-\mu^{ads}}{T}\right), \end{array}$$(36)$$\begin{array}{ccc} J_{net} & = & \frac{k_{a}}{R}a^{g}\left(\frac{\mu^{g}-\mu^{g\phi }}{T}\right). \end{array}$$The two expressions, Equations (Equation 35) and (36), are equivalent but with the reference in the adsorbed phase and in the gas, respectively. Through comparison with Equations (Equation 30) and (31), under isothermal conditions, we obtain the following directly: (37)$$L_{\mu \mu }^{ads}=L_{\mu \mu }^{g}=k_{a}a^{g}/R=k_{d}a_{eq}^{ads}/R.$$It is interesting to note in the equation above how the mass transfer coefficient is written in terms of a product involving the rate constant and the activity. In other words, it is described by the kinetic and thermodynamic aspects of the process.

Far from equilibrium, Equations (33) and (34) have to be used without approximations. Thus, the expression of $L_{\mu \mu }$ shows dependence on the difference in the chemical potential between the actual value and the value in equilibrium. From Equations (33) and (34), we find the following:(38)$$\begin{array}{ccc} L_{\mu \mu }^{ads,ext} & = & L_{\mu \mu }^{ads}\left[\frac{1-\exp\left(-X\right)}{X}\right],\;\mathrm{with}\;X=\frac{\mu_{eq}^{ads}-\mu^{ads}}{RT}, \end{array}$$(39)$$\begin{array}{ccc} L_{\mu \mu }^{g,ext} & = & L_{\mu \mu }^{g}\left[\frac{1-\exp\left(-Z\right)}{Z}\right],\;\mathrm{with}\;Z=\frac{\mu^{g}-\mu^{g\phi }}{RT}. \end{array}$$
where the superscript ext indicates the properties extending beyond close equilibrium. Obviously, Z=X and $L_{\mu \mu }^{ads,ext}=L_{\mu \mu }^{g,ext}$ since $L_{\mu \mu }^{ads}=L_{\mu \mu }^{g}$. It is worth noting that the first corrective term for $L_{\mu \mu }^{ads}$, using Taylor’s expansion, is proportional to *X*: $L_{\mu \mu }^{g,ext}\approx L_{\mu \mu }^{g}\left(1+X/2\right)$.

#### 2.2.2. Non-Isothermal Conditions

In this part, the temperature of both phases, gas, $T^{g}$, and adsorbed, $T^{ads}$, differs but is kept constant during the adsorption process. The final state is a stationary non-equilibrium situation characterised by constant non-zero heat flux and zero mass flux. At zero mas flux, $J_{net}=0$, $k_{a}\left(T^{g}\right)a^{g}\left(T^{g}\right)=k_{d}\left(T^{ads}\right)a_{stat}^{ads}\left(T^{ads}\right)$, and an equivalent of the Langmuir adsorption isotherm for non-isothermal conditions can be written using Equations (8) and (9):(40)$$\begin{array}{ccc} \theta_{stat}\left(T^{ads}\right) & = & \frac{K_{NI}p\left(T^{g}\right)/p^{0}}{1+K_{NI}p\left(T^{g}\right)/p^{0}}, \end{array}$$(41)$$\begin{array}{ccc} K_{NI}\left(T^{g},T^{ads}\right) & = & k_{a}\left(T^{g}\right)/k_{d}\left(T^{ads}\right)=1.94\times 10^{-4}\exp\left(\frac{1500}{RT^{g}}+\frac{4400}{RT^{ads}}\right), \end{array}$$
where $K_{NI}$ is the non-isothermal Langmuir constant. In this expression, the subscript stat is used to identify the final stationary state.

For clarity in the following part, the temperature dependence of the kinetic parameters and thermodynamic properties will be omitted. We first introduce $a_{stat}^{g\phi }$, which refers to the activity of the gas phase in the stationary state with the actual adsorbed phase at a temperature of $T^{ads}$ using Equation (41). This is used to defined two sets of transport equations, like for Equations (Equation 30) and (31). From Equations (Equation 4), (5), and (Equation 16),(42)$$\begin{array}{ccc} J_{net} & = & k_{d}\left(a_{stat}^{ads}-a^{ads}\right)=k_{a}\left(a^{g}-a_{stat}^{g\phi }\right), \end{array}$$(43)$$\begin{array}{ccc} J_{net} & = & k_{d}\exp\left(\frac{-\mu^{★}+\mu_{stat}^{ads}}{RT^{ads}}\right)\left[1-\exp\left(\frac{\mu^{ads}-\mu_{stat}^{ads}}{RT^{ads}}\right)\right], \end{array}$$(44)$$\begin{array}{ccc} J_{net} & = & k_{a}\exp\left(\frac{-\mu^{0}+\mu^{g}}{RT^{g}}\right)\left[1-\exp\left(\frac{\mu^{g\phi }-\mu^{g}}{RT^{g}}\right)\right]. \end{array}$$Approaching the limit of the stationary state, the net mass flux can be written in a similar way to that for the isothermal case: (45)$$\begin{array}{ccc} J_{net} & = & \frac{k_{d}}{R}a_{stat}^{ads}\left(\frac{\mu_{stat}^{ads}-\mu^{ads}}{T^{ads}}\right), \end{array}$$(46)$$\begin{array}{ccc} J_{net} & = & \frac{k_{a}}{R}a^{g}\left(\frac{\mu^{g}-\mu^{g\phi }}{T^{g}}\right). \end{array}$$This expression is very similar to the expression for the isothermal case, Equations (Equation 35) and (36). However, it does not make a clear distinction between the contribution of the temperature and the difference in the chemical potential, like in Equations (Equation 30) and (31). This can be achieved for the adsorbed phase reference by rewriting Equation (Equation 42):(47)$$\begin{array}{cc} J_{net} & =k_{d}\left(T^{g}\right)a_{eq}^{ads}\left(T^{g}\right)-k_{d}\left(T^{ads}\right)a^{ads}\left(T^{ads}\right), \end{array}$$(48)$$\begin{array}{cc} = & k_{d}\left(T^{ads}\right)\exp\left[\frac{-4400}{R}\left(\frac{1}{T^{g}}-\frac{1}{T^{ads}}\right)\right]a_{eq}^{ads}\left(T^{g}\right)-k_{d}\left(T^{ads}\right)a^{ads}\left(T^{ads}\right), \end{array}$$(49)$$\begin{array}{cc} = & k_{d}\exp\left(\frac{-\mu^{★}+\mu_{eq}^{ads}}{RT^{ads}}\right)\left\{\exp\left[\frac{-4400}{R}\left(\frac{1}{T^{g}}-\frac{1}{T^{ads}}\right)\right]-\exp\left(\frac{\mu^{ads}-\mu_{eq}^{ads}}{RT^{ads}}\right)\right\}. \end{array}$$In Equation (49), all of the chemical potentials are given for a temperature of $T^{ads}$, and we use the equilibrium condition $a_{eq}^{ads}\left(T^{g}\right)=a_{eq}^{ads}\left(T^{ads}\right)$:(50)$$a_{eq}^{ads}\left(T\right)=\frac{\theta_{eq}}{1-\theta_{eq}}=\exp\left(\frac{-\mu^{★}\left(T\right)+\mu_{eq}^{ads}\left(T\right)}{RT}\right).$$
From Equation (Equation 42), the net flux can alternatively be written taking the gas phase reference:(51)$$\begin{array}{ccc} J_{net} & = & k_{a}\left(T^{g}\right)a^{g}\left(T^{g}\right)-k_{a}\left(T^{ads}\right)a^{g\phi }\left(T^{ads}\right), \end{array}$$(52)$$\begin{array}{cc} = & k_{a}\left(T^{g}\right)\left\{a^{g}\left(T^{g}\right)-\exp\left[\frac{-1500}{R}\left(\frac{1}{T^{g}}-\frac{1}{T^{ads}}\right)\right]a^{g\phi }\left(T^{ads}\right)\right\}, \end{array}$$(53)$$\begin{array}{cc} = & k_{a}\exp\left(\frac{-\mu^{0}+\mu^{g\phi }}{RT^{g}}\right)\left\{\exp\left(\frac{\mu^{g}-\mu^{g\phi }}{RT^{g}}\right)-\exp\left[\frac{-1500}{R}\left(\frac{1}{T^{g}}-\frac{1}{T^{ads}}\right)\right]\right\} \end{array}$$
where we again used $a^{g\phi }\left(T^{ads}\right)=a^{g\phi }\left(T^{g}\right)$.

Approaching the stationary state and imposing the concept of $T^{g}$ and $T^{ads}$ being close, Equations (49) and (53) can be expressed:(54)$$\begin{array}{ccc} J_{net} & = & k_{d}a_{eq}^{ads}\left[\frac{-4400}{R}\left(\frac{1}{T^{g}}-\frac{1}{T^{ads}}\right)+\left(\frac{\mu_{eq}^{ads}-\mu^{ads}}{RT^{ads}}\right)\right], \end{array}$$(55)$$\begin{array}{ccc} J_{net} & = & k_{a}a^{g\phi }\left[\frac{1500}{R}\left(\frac{1}{T^{g}}-\frac{1}{T^{ads}}\right)+\left(\frac{\mu^{g}-\mu^{g\phi }}{RT^{g}}\right)\right]. \end{array}$$Through comparison with Equations (Equation 30) and (31), we obtain the following: (56)$$\begin{array}{ccc} L_{\mu q}^{ads} & = & k_{d}a_{eq}^{ads}\frac{4400}{R}=4400L_{\mu \mu }^{ads}, \end{array}$$(57)$$\begin{array}{ccc} L_{\mu q}^{g} & = & -k_{a}a^{g\phi }\frac{1500}{R}=-1500L_{\mu \mu }^{g}. \end{array}$$$L_{\mu \mu }^{ads}$ and $L_{\mu \mu }^{g}$ are expressed the same way as they are for the isothermal case but they are not equal since the temperature difference is not null.

Another transport quantity that quantifies the coupling between the heat and mass fluxes is the heat of transfer [9,10], q∗. This gives the heat transported with the mass flux at a constant temperature. Bearing in mind the aforementioned transport coefficients, it is defined according to the equations below: (58)$$\begin{array}{ccc} q^{*ads} & = & {\left(\frac{J_{q}^{ads}}{J_{net}}\right)}_{\left(T^{ads}=T^{g}\right)}=\frac{L_{\mu q}^{ads}}{L_{\mu \mu }^{ads}}=4400\;J/\mathrm{mol}, \end{array}$$(59)$$\begin{array}{ccc} q^{*g} & = & {\left(\frac{J_{q}^{g}}{J_{net}}\right)}_{\left(T^{ads}=T^{g}\right)}=\frac{L_{\mu q}^{g}}{L_{\mu \mu }^{g}}=-1500\;J/\mathrm{mol}. \end{array}$$Here, $J_{q}^{ads}$ and $J_{q}^{g}$ are the measurable heat fluxes on the adsorbed side and on the gas side of the surface buffer. $q^{*ads}$ and $q^{*g}$ are constant in our non-Langmuir kinetics model and independent of temperature. Similar behaviour has been observed for the direct computation of $q^{*}$ from molecular dynamics simulations of the adsorption of n-butane onto zeolite silicalite [11]; it is interesting to note that this is not the case for a liquid–vapour interface [10]. As expected [10], the difference $q^{*g}-q^{*ads}$ gives the enthalpy of adsorption, being $\Delta_{ads}H=-5900\;J/\mathrm{mol}$ here.

## 3. Results and Discussion

### 3.1. Isothermal Kinetics of Adsorption and Desorption

Equation (Equation 23) was used to investigate the isothermal kinetics of adsorption and desorption. We used a pressure of 50 bar and temperatures of 77 K and 293 K. These are the same conditions as those reported by Burress et al. [12]. Through applying Langmuir’s isotherm equation, Equation (Equation 1), the equilibrium coverages are 0.990 and 0.0987 for 77 and 293 K, respectively.

Figure 1 reports the result at 293 K. The coverage, θ, and the logarithmic function of $\left|\theta_{eq}-\theta \right|$ are plotted as a function of time according to Equation (Equation 23) for adsorption and desorption. Logarithmic behaviour is observed. The characteristic time is computed from Equation (Equation 25), and the results are 4.3 and 4.6 ps for adsorption and desorption, respectively. These results agree well with the characteristic time of 4.9 ps found using the Henry’s regime expression, Equation (Equation 29). This is expected since the range of coverage is small, lower than 0.1, where Henry’s regime is supposed to apply. They are also in the same order of magnitude as those in Burress et al. [12] (16 ps for adsorption and 2.1 ps for desorption), although these authors used Langmuir dynamics.

At 77 K, the equilibrium coverage corresponding to 50 bar is $\theta_{eq}=0.990$, close to saturation. The kinetics of desorption/adsorption from 0 to $\theta_{eq}$ are calculated using Equation (Equation 23). The results are given in Figure 2 and Figure 3. The adsorption curve is linear below a coverage of 0.95 and tends towards logarithmic behaviour beyond this point. For desorption, the logarithmic behaviour is spread over a wider coverage range; it is observed below a coverage of 0.2. We obtain a characteristic desorption time of 360 ps using Equation (Equation 25), which is much longer than the value for adsorption, at about 6.2 ps. This difference can be explained by the asymmetry in the values of the activity of the gas and adsorbed phases, which results in an asymmetric net flux, as displayed for 77 K in Figure 4. During adsorption, the net flux is at the maximum for most of the coverage, below 0.9, while it is nearly null for desorption in the same range. The desorption kinetics are then much less efficient than the adsorption kinetics. Burress et al. [12], for the same conditions, gave characteristic times about two times higher, at 15 ps for adsorption and 530 ps for desorption. Nonetheless, the model presents the same qualitative features, as the estimated times are in the same order of magnitude and the strong asymmetry between the adsorption and desorption kinetics is reproduced.

The two previous simulations were carried out under Henry’s regime and near saturation. The results are reported in Figure 5 for intermediate coverage. At 293 K, the adsorption and desorption from 50 to 450 bar were studied. From Langmuir’s isotherm equation, Equation (Equation 1), the equilibrium coverages are 0.0987 and 0.496. Although this range is rather wide, about 0.4, the logarithmic domain appears rapidly after 3 ps. The characteristic time τ is 1.9 ps for adsorption and 2.9 ps for desorption, and these values are smaller than that for Henry’s regime (4.9 ps) presented above.

In order to quantify the adsorption kinetics close to equilibrium, the transport coefficients $L_{\mu \mu }$ were computed for four different temperatures using Equation (Equation 37). The results are plotted in Figure 6 for 77 K, 273 K, 293 K, and 313 K. The values of $L_{\mu \mu }$ show an increase as a function of both the coverage and pressure. For constant coverage, the values of $L_{\mu \mu }$ increase with temperature, but the trend is the reverse at a constant pressure. These trends are in agreement with those observed in non-equilibrium molecular dynamics studies of the adsorption of *n*-butane onto zeolite silicalite [11]. A similar dependence of $L_{\mu \mu }$ (or inversely that for resistance) on pressure and temperature has also been observed at liquid–vapour interfaces both in molecular simulations and using theoretical models [10,13,14,15,16,17].

Beyond close equilibrium, the transport coefficients, $L_{\mu \mu }^{ext}$, are computed using Equation (Equation 38). In Figure 7, the values for adsorption between 0 and 50 and 50 and 450 bar and for desorption between 450 and 50 bar at 293 K are reported. They are compared with the reference $L_{\mu \mu }$ values. As expected, $L_{\mu \mu }$ and $L_{\mu \mu }^{ext}$ agree close to equilibrium, but they differ significantly outside of this condition. With the same equilibrium values, the desorption between 450 and 50 bar and the adsorption between 0 and 50 bar show continuous linear evolution of the values of $L_{\mu \mu }^{ext}$ around equilibrium. The slope of $L_{\mu \mu }$ with coverage is steeper than that of $L_{\mu \mu }^{ext}$, which indicates closer values to those for equilibrium for the latter. This is important since in general, it is common to consider constant transfer coefficients during the kinetics. From the results, this can be considered to be a good approximation close to the equilibrium state. The asymmetry between the adsorption and desorption kinetics that was mentioned above is also evidenced by comparing the values of $L_{\mu \mu }^{ext}$ during the adsorption and desorption kinetics from 50 to 450 bar.

### 3.2. Non-Isothermal Kinetics of Adsorption

So far, Equation (Equation 23) has been derived to account for the evolution of θ under isothermal conditions. It is also possible to derive this kind of kinetic equation for non-isothermal conditions with both phases constant but different temperatures and the gas phase at a constant pressure. The final expression is the same as that for Equation (Equation 23) but uses the non-isothermal Langmuir constant $K_{NI}$ in the derivation—see Equation (41)—instead of the equilibrium Langmuir constant. The non-isothermal kinetics of adsorption are presented in Figure 8. A 450-bar-pressure gas phase is maintained at 293 K, and different surface temperatures are considered: 273 K, 283 K, 293 K, 303 K, and 313 K. As described in Section 2.2.2, the final stationary state is outside of equilibrium and characterised by zero net mass flux and non-zero heat flux. The adsorption kinetics are very similar to those in isothermal conditions, and as expected, the final coverage decreases as the surface temperature increases.

In the stationary state, the coverage can be obtained directly from Equation (41) under isothermal and non-isothermal conditions. The results are compared in Figure 9 for surface temperatures ranging from 73 to 503 K and using a constant $T^{g}=293$ K for the non-isothermal case. The curves exhibits a similar inverse S shape, with a crossing point at the temperature of 293 K. Below this temperature, the non-isothermal coverage is lower than the equilibrium value, and above this point, it is higher; this is easily understandable according to Le Chatelier’s principle. The difference between these two curves does not exceed 0.03 in coverage, which, except when approaching zero coverage, can be considered negligible in most cases or at least a secondary effect. This is also the case for the kinetics. It is important to note that because the adsorption is exothermic, strong non-isothermal effects can occur and delay the kinetics of adsorption/desorption. For example, in reference [18], Inzoli et al. have evidenced, using molecular dynamics, that the long-term adsorption kinetics of *n*-butane on silicalite are governed by the heat of the transfer between the zeolite and the gas phase. This is clearly due to the exothermic effect of adsorption.

## 4. Conclusions

The adsorption and desorption kinetics of H_2_ on graphite are studied based on the non-Langmuir adsorption and desorption equilibrium rates obtained in a previous article [7]. New analytical expressions are derived to model the kinetics both under isothermal and non-isothermal conditions. The results show good agreement with the data given by Burress et al. [12]. In particular, the asymmetry in time between the adsorption and the desorption is pointed out, which is due to the asymmetry in the thermodynamic forces between adsorption and desorption.

A link with the transport equations for the non-equilibrium thermodynamics of heterogeneous systems is proposed, and expressions for the transport coefficients $L_{\mu \mu }$ and $L_{\mu q}$ and the heat of transport are also given. They are written as a product of the dynamical and thermodynamical properties. The results are in qualitative agreement with previous studies on the transport of mass across interfaces from molecular dynamics simulations. The values of $L_{\mu \mu }$ increase with the coverage and temperature.

These equations offer new additional tools that can be used for a better analysis and understanding of the kinetics of adsorption obtained both in experiments and molecular simulations.

## Figures and Tables

**Figure 1 entropy-27-00229-f001:**
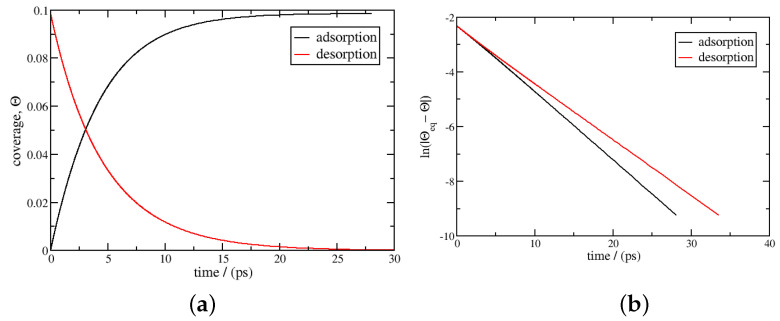
Adsorption and desorption kinetics using Equation (Equation 23); the temperature is 293 K, with the pressure between 0 and 50 bar at a temperature of 293 K. The equilibrium coverages are 0 and 0.0987, respectively. In the left figure (**a**), the coverage θ is plotted as a function of time, *t*; in the right figure (**b**), $ln(\left|\theta_{eq}-\theta \right|)$ is plotted as a function of time.

**Figure 2 entropy-27-00229-f002:**
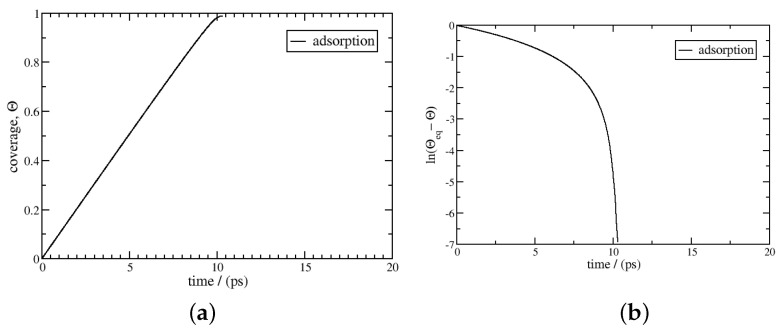
Adsorption kinetics using Equation (Equation 23); the temperature is 77 K, and the initial coverage and the coverage at equilibrium are 0 and 0.990, respectively, at a pressure of 50 bar. In the left figure (**a**), the coverage θ is plotted as a function of time, *t*; in the right figure (**b**), $ln\left(\right|\theta -\theta_{eq}\left|\right)$ is plotted as a function of time. The kinetics do not follow logarithmic behaviour in the coverage range of the plot. The characteristic time is 6.2 ps.

**Figure 3 entropy-27-00229-f003:**
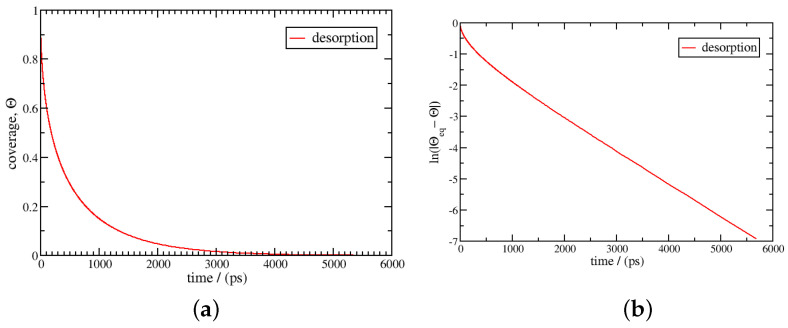
Desorption kinetics using Equation (Equation 23); the temperature is 77 K, and the initial coverage and the coverage at equilibrium are 0.990 and 0, respectively. In the left figure (**a**), the coverage θ is plotted as a function of time, *t*; in the right figure (**b**), $ln\left(\right|\theta -\theta_{eq}\left|\right)$ is plotted as a function of time. After 50 ps, the kinetics follow logarithmic behaviour, and the characteristic time scale for desorption is 360 ps.

**Figure 4 entropy-27-00229-f004:**
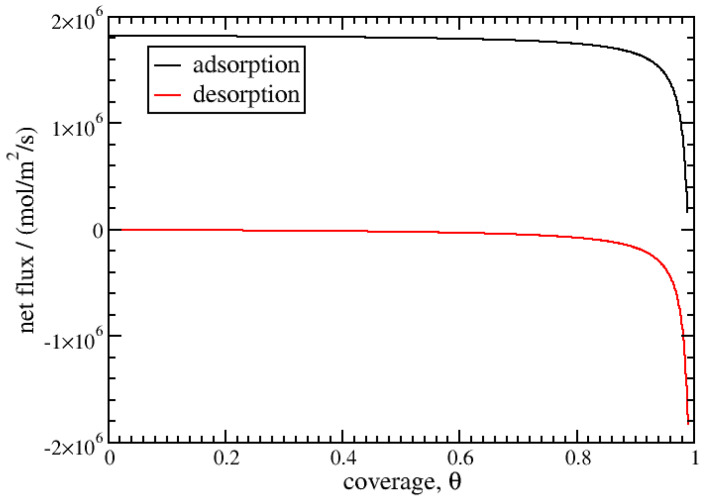
Evolution of the net flux for desorption and adsorption as a function of the coverage at 77 K between a coverage of 0 and 0.990.

**Figure 5 entropy-27-00229-f005:**
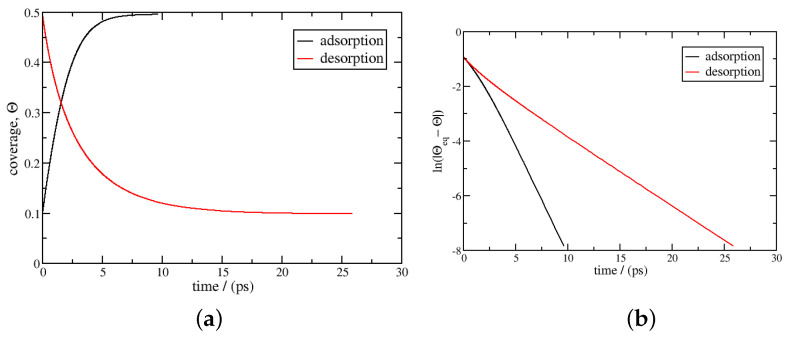
Adsorption and desorption kinetics using Equation (Equation 23); the pressure ranges between 50 and 450 bar at a temperature of 293 K. The equilibrium coverages are 0.0987 and 0.496, respectively. In the left figure (**a**), the coverage θ is plotted as a function of time, *t*; in the right figure (**b**), $ln\left(\right|\theta_{eq}-\theta \left|\right)$ is plotted as a function of time. The characteristic time scales for adsorption and desorption are 1.9 ps and 2.9 ps.

**Figure 6 entropy-27-00229-f006:**
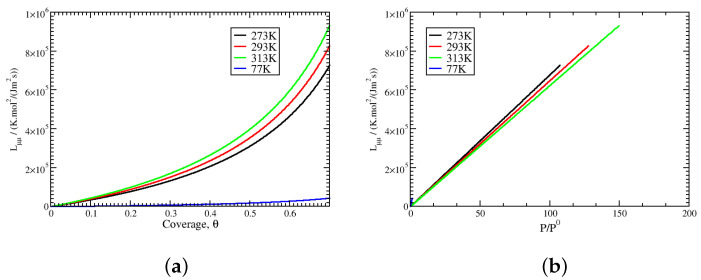
Mass transport coefficient $L_{\mu \mu }$ as a function of the coverage (**a**) and pressure (**b**) at 77 K, 273 K, 293 K, and 313 K.

**Figure 7 entropy-27-00229-f007:**
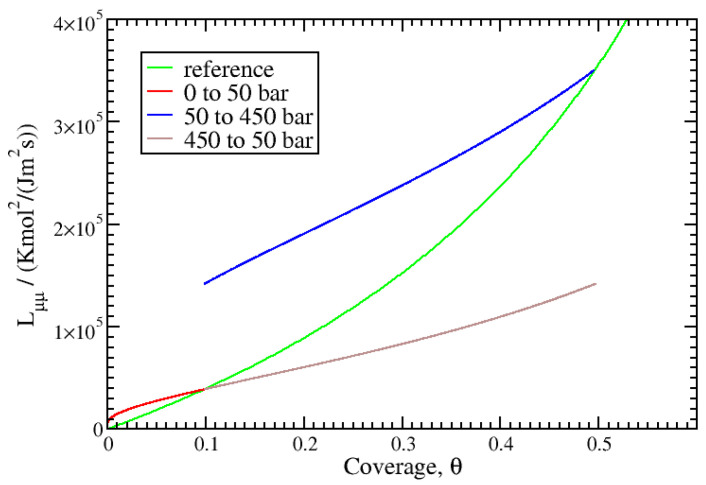
Comparison between the mass transport coefficients $L_{\mu \mu }$ obtained close to equilibrium at 293 K and $L_{\mu \mu }^{ext}$ during adsorption or desorption calculated using Equation (Equation 38).

**Figure 8 entropy-27-00229-f008:**
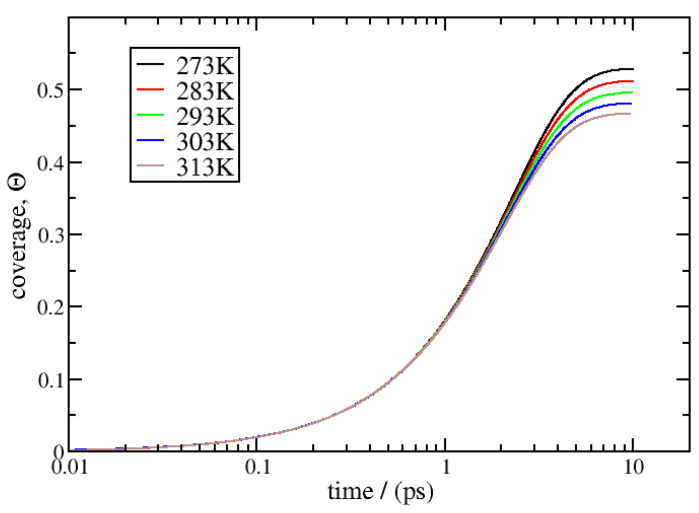
Non-isothermal adsorption kinetics calculated from Equation (Equation 23) using an initial bare surface and a constant gas pressure of 450 bar at 293 K with surface temperatures of 273 K, 283 K, 293 K, 303 K, and 313 K. The final state is outside of equilibrium.

**Figure 9 entropy-27-00229-f009:**
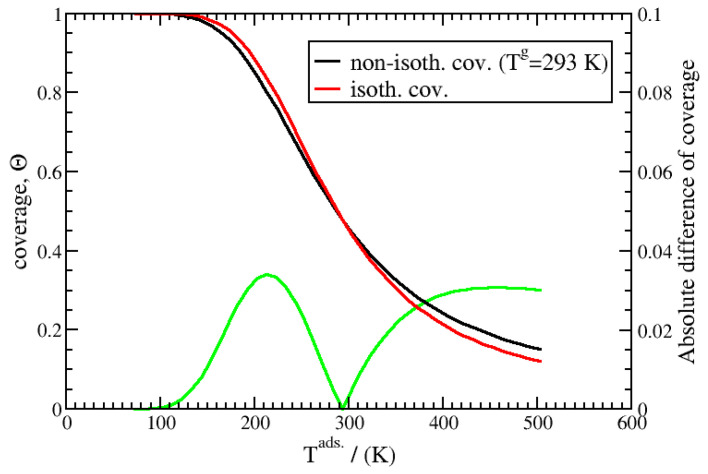
Comparison between stationary coverage under isothermal and non-isothermal conditions using Equation (41) and a constant gas pressure of 50 bar at 293 K for non-isothermal situations. The absolute difference between the two curves for each temperature is given in green.

## Data Availability

Data are the results of the equations provided within the article.

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
