# Peer review of "Adsorption Kinetics Model of Hydrogen on Graphite"

_entropy, 2025, doi:10.3390/e27030229_

Round 1

Reviewer 1 Report

Comments and Suggestions for Authors

See attached 

Reviewer 2 Report

Comments and Suggestions for Authors

Adsorption thermodynamics  have been extensively studied and a large number of models describe different situations. The adsorption kinetics has been less addressed from a theoretical point of view due to its complexity. This work attempts to attack this problem by developing macroscopic models to describe adsorption and desorption kinetics using irreversible thermodynamics. Specifically, new analytical expressions are derived to model the kinetics under both isothermal and non-isothermal conditions. The results show good agreement with experimental data.

The manuscript is very well structured and written, meets the proposed objectives and I believe it will be of great interest to the scientific community and, in particular, to the readers of the journal "Entropy".

I did not find any aspect in the manuscript that I consider should be improved before the manuscript is published.

Author Response

We thank the reviewer for his/her nice comments. We hope, if accepted for publication, that  the readers will be as enthusiastic as the two reviewers.

Reviewer 3 Report

Comments and Suggestions for Authors

Authors present the derivation of a new formalism to describe adsorption and desorption of Hon graphite under isothermal and non-isothermal conditions. The connections with the Langmuir kinetics and the Henry's regime are established before to treat the transport equations within isothermal and non-isothermal regimes.

I have enjoyed reading the present manuscript, which offers an extense introduction where the overall problematic treated in this study is exposed. The formalism is also well discussed and developed, and despite the extension of the text, it does not come up difficult to follow. 

The only criticism I would raise here has to do with the figures. They are hard to see and seem to be the result of a not too elaborated procedure: insets which sometimes do not fit in the area covered by the figure -Fig 9-, labels which could use scientific format - Figures 4 and 7-, etc...

In pg. 9 line 263 it is said that the equilibrium coverages are 0.0987 and 0.990 at 77 and 293 K, respectively, but a bit later in page 10 line 274, the authors say: "At 77 K the equilibrium coverage corresponding to 50 bar is θeq = 0.990, close to saturation". Isn't there here a contradiction? Is there a mistake?

Author Response

The authors thank the reviewer for the nice comments and suggestions for improving the manuscript. 

Comment 1: 

The only criticism I would raise here has to do with the figures. They are hard to see and seem to be the result of a not too elaborated procedure: insets which sometimes do not fit in the area covered by the figure -Fig 9-, labels which could use scientific format - Figures 4 and 7-, etc...

Answer:

The format of the figures has been changed, they are, from our point of view, more easy to read. Figure 9  has been changed and should follow now the requirement of the reviewer. For the label format all figures are now mention by Figure in the text, it is also a suggestion from the template provided by the Journal.

Comment 2:

In pg. 9 line 263 it is said that the equilibrium coverages are 0.0987 and 0.990 at 77 and 293 K, respectively, but a bit later in page 10 line 274, the authors say: "At 77 K the equilibrium coverage corresponding to 50 bar is θeq = 0.990, close to saturation". Isn't there here a contradiction? Is there a mistake?

Answer: We thank the reviewer for this question. This is clearly a mistake, it has been corrected in the text: " 

the equilibrium coverages are 0.990 and 0.0987, for 77 and 293 K respectively." around line 264, the text is writen in color red in the manuscript.

We hope these modifications will satisfy the reviewer. 

Round 2

Reviewer 1 Report

Comments and Suggestions for Authors

 I am completely satisfied with the authors' reply. Moreover, I would like to apologize for overlooking the main point related to non-Langmuir kinetics.